# Dual LSD1 and HDAC6 Inhibition Induces Doxorubicin Sensitivity in Acute Myeloid Leukemia Cells

**DOI:** 10.3390/cancers14236014

**Published:** 2022-12-06

**Authors:** Ipek Bulut, Adam Lee, Buse Cevatemre, Dusan Ruzic, Roman Belle, Akane Kawamura, Sheraz Gul, Katarina Nikolic, A. Ganesan, Ceyda Acilan

**Affiliations:** 1Graduate School of Health Sciences, Koc University, Sariyer 34450, Turkey; 2School of Pharmacy, University of East Anglia, Norwich NR4 7TJ, UK; 3Research Center for Translational Medicine (KUTTAM), Koc University, Sariyer 34450, Turkey; 4Department of Pharmaceutical Chemistry, Faculty of Pharmacy, University of Belgrade, 11000 Belgrade, Serbia; 5Chemistry Research Laboratory, University of Oxford, Mansfield Road, Oxford OX1 3TA, UK; 6Chemistry—School of Natural and Environmental Sciences, Newcastle University, Bedson Building, Kings Road, Newcastle Upon Tyne NE1 7RU, UK; 7Fraunhofer Institute for Translational Medicine and Pharmacology ITMP, 22525 Hamburg, Germany; 8Fraunhofer Cluster of Excellence for Immune-Mediated Diseases CIMD, 22525 Hamburg, Germany; 9Department of Medical Biology, School of Medicine, Koc University, Sariyer 34450, Turkey

**Keywords:** epigenetics, histone deacetylases (HDACs), lysine demethylases (KMDs), multitargeting, combination therapy, acute myeloid leukemia

## Abstract

**Simple Summary:**

GSK2879552 is a LSD1 inhibitor in clinical development. By structural modification, we obtained an analogue that is a potent and selective dual inhibitor of HDAC6 and LSD1 (IC_50_ 110 and 540 nM, respectively). The dual targeting agent was superior to GSK2879552 in the growth inhibition of two acute myeloid leukemia (AML) cell lines. In combination experiments, the dual inhibitor primed AML cells to apoptosis with a sublethal concentration of doxorubicin. Our data suggest that doxorubicin toxicity can be reduced by parallel inhibition of HDAC6 and LSD1.

**Abstract:**

Defects in epigenetic pathways are key drivers of oncogenic cell proliferation. We developed a LSD1/HDAC6 multitargeting inhibitor (iDual), a hydroxamic acid analogue of the clinical candidate LSD1 inhibitor GSK2879552. iDual inhibits both targets with IC_50_ values of 540, 110, and 290 nM, respectively, against LSD1, HDAC6, and HDAC8. We compared its activity to structurally similar control probes that act by HDAC or LSD1 inhibition alone, as well as an inactive null compound. iDual inhibited the growth of leukemia cell lines at a higher level than GSK2879552 with micromolar IC_50_ values. Dual engagement with LSD1 and HDAC6 was supported by dose dependent increases in substrate levels, biomarkers, and cellular thermal shift assay. Both histone methylation and acetylation of tubulin were increased, while acetylated histone levels were only mildly affected, indicating selectivity for HDAC6. Downstream gene expression (CD11b, CD86, p21) was also elevated in response to iDual treatment. Remarkably, iDual synergized with doxorubicin, triggering significant levels of apoptosis with a sublethal concentration of the drug. While mechanistic studies did not reveal changes in DNA repair or drug efflux pathways, the expression of AGPAT9, ALOX5, BTG1, HIPK2, IFI44L, and LRP1, previously implicated in doxorubicin sensitivity, was significantly elevated.

## 1. Introduction

The enzymes that introduce or remove epigenetic modifications in DNA and histone proteins, as well as the binding domains that recognize these marks, modulate transcriptional activation or repression. Dysregulation of these events is strongly linked to oncogenesis and cancer progression and can be corrected by small molecule inhibitors. Epigenetic reprogramming by such compounds is a relatively recent addition to cancer chemotherapy [1,2]. Within the last two decades, epigenetic drug discovery (EPIDD) has successfully produced approximately 100 clinical candidate epi-drugs, resulting in eight approvals [3,4].

Histone acetylation and methylation are two major epigenetic marks that are removed by erasers, including the zinc-dependent histone deacetylases (HDACs) and the FAD-dependent lysine-specific demethylase 1 (LSD1, also known as KDM1A) that are both targets for EPIDD [5,6]. While LSD1 has a single well validated substrate in histone H3 [7], the eleven human HDAC isozymes are highly diverse in their function [8]. HDAC1, HDAC2, and HDAC3 exist in multiprotein transcriptional complexes that deacetylate histones and other nuclear proteins and are truly epigenetic in their physiological role. HDAC6, on the other hand, predominantly deacetylates cytoplasmic substrates. Of the remaining isozymes, HDAC10 is a polyamine deacetylase, HDAC8 and HDAC11 are long chain fatty acid deacylases, and HDAC4, HDAC5, HDAC7, and HDAC9 are pseudoenzymes with low catalytic activity.

Both HDACs and LSD1 are highly expressed in various cancer types, and LSD1 is itself a substrate for HDAC1-mediated deacetylation at Lys374, which influences histone binding [9]. HDAC1, HDAC2, and LSD1 are found together in transcriptional repressor complexes, such as NuRD, CoREST, and Sin 3A, which promote cancer cell survival [10]. Furthermore, the co-administration of HDAC and LSD1 inhibitors in cancer cell lines was demonstrated to be synergistic in vitro [11]. We were interested in achieving the same effects through a single agent capable of interacting with both enzymes. Such multitargeting inhibitors are increasingly popular in EPIDD, as they avoid the pharmacokinetic complexities of drug cocktails in combination therapy, and several examples have advanced to clinical trials [12,13]. We aimed for selective inhibition of the HDAC6 isozyme, as its cytoplasmic localization would reduce the global transcriptional side effects that limit the tolerability of inhibitors that target the nuclear HDAC1, HDAC2, and HDAC3 isozymes. Finally, we investigated combinations of the dual HDAC6/LSD1 inhibitor with cytotoxic agents that are the current standard of care in acute myeloid leukemia (AML) and discovered a significant enhancement of doxorubicin sensitivity.

## 2. Materials and Methods

### 2.1. Inhibitor Synthesis

By modifying the industrial route towards the LSD1 inhibitor GSK2879552 (Cayman Chemical, 19403), we obtained the analogue iDual in which the carboxylic acid is replaced by a hydroxamic acid, a well-known metal binding group and a common motif within HDAC inhibitors. As additional control compounds, we prepared the isomeric iHDAC6 and a null compound iNC in which the altered position of the phenyl ring within the cyclopropyl ring results in loss of activity against LSD1. Full details of the synthesis and the characterization data are provided in the Appendix A (Appendix A).

### 2.2. LSD1 MALDI-TOF Enzyme Assay

Recombinant human LSD1 was prepared, as previously described [14]. Test compounds were dissolved in DMSO to 10 mM and dispensed (100 nL, 2.5 nL increments) at appropriate quantities on 384-well AlphaScreen ProxiPlates™ (PerkinElmer, Boston, MA, USA) using Labcyte Echo 550^®^ dispenser (Labcyte, Sunnyvale, CA, USA). Dilution series were made using Multidrop™ (ThermoFisher, Waltham, MA, USA), and samples were backfilled with DMSO (up to 100 nL), if necessary. Enzymatic reactions were conducted in fresh prepared assay buffer [Triton™ X-100 (Sigma Chemical Co, St Louis, MO, USA) (0.01%), (tris(2-carboxyethyl) phosphine·hydrochloride salt (TCEP, 2.0 mM), Tris (20 mM) in MilliQ (Millipore, Bedford, MA, USA), (pH 8.0)]. Enzyme, peptide, and quench solutions were added to the wells using E1-ClipTip™ (ThermoFisher) with corresponding E1-ClipTip pipette tips. Preparation of the solutions and enzymatic reactions were conducted at room temperature. The enzyme (5.0 µL of a 0.3 µM solution in assay buffer) was added to the prepared inhibitor plate, and the mixture was incubated for 10 min. The peptide substrate histone H3 1-21K4me (5.0 µL of a 10.0 µM solution in assay buffer) was added to the mixture and the plate was further incubated for 10 min, followed by quenching with formic acid (5.0 µL of a 1% solution in MilliQ water). The samples were spotted (1.0 µL) on a MALDI-TOF target plate. MALDI-TOF matrix saturated solution of α-cyano-4-hydroxy-cinnamic acid (Aldrich, St Quentin Fallavier, France) (10 mg/mL) of (1.0 µL) was added, mixed, and air dried at room temperature. Data were acquired using Bruker Daltonics AutoFlexSpeed MALDI-TOF ( Bruker Daltonics, Bremen, Germany) in positive reflectron mode. The instrument was operated using the software Compass for Flex series (v. 1.4) and Flex control (v. 3.4). Acquired data (Appendix A) were processed using Flex analysis (v. 3.4) (Bruker Daltonics), Microsoft Excel™ (2010) (Microsoft, California, CA, USA), GraphPad Prism^®^ (v. 5.04) (GraphPad Software, San Diego, CA, USA), and images were made using Adobe^®^ Illustrator^®^ (v. 15.0) (Adobe Systems, San Jose, CA, USA).Experiments were performed in triplicate.

### 2.3. HDAC Enzyme Assay

Human recombinant C-ter-His-FLAG-HDAC1 (50051), HDAC-3/NcoR2 (50003), N-ter-GST-HDAC-6 (50006), and C-terminal His-tag-HDAC8 (50008) proteins were purchased from BPS Bioscience (San Diego, CA, USA). Trichostatin A (Sigma-Aldrich, St. Louis, MO, USA) dissolved in DMSO (starting concentration 2 µM) was used as a positive control, whereas DMSO was used as a negative control. Synthesized compounds were dissolved in DMSO (*v/v*) and stored at −20 °C. Inhibition profiles for the synthesized compounds were monitored with the bioluminogenic HDAC-Glo™ I/II assay (Promega Corp, Madison, WI, USA), following protocols in the literature [15]. The compounds were tested at 11 different concentrations with 2-fold dilutions, starting from 10 µM in HDAC6 and HDAC8 enzyme assays and 100 µM in HDAC1 and HDAC3 enzyme assays. Compound solutions were dispensed using the Echo 550^®^ (Labcyte, Sunnyvale, CA, USA) into 384 well assay plates (10 µL/well). This was followed by the addition of HDAC enzyme (5 µL/well) using the Multidrop™ (Thermo, Waltham, MA, USA) and incubation for 10 min at room temperature. The final addition of the HDAC-Glo assay reagent (10 μL per well) initiated the luciferase reaction. After 10 min incubation at room temperature, the luminescence was read with an EnSpire Microplate Reader (PerkinElmer, Waltham, MA, USA). All the experiments were performed in triplicate, and the raw data obtained after screening were analyzed using Prism Software (GraphPad Software v. 6). The dose response curves (Appendix A) were generated using a 4-parameter logistic fit in 11-point format yielding the IC_50_ values.

### 2.4. Cell Lines

ATCC derived cell lines: THP-1 (TIB-202), Jurkat (TIB-152), K-562 (CCL-243), HEK293-T (CRL-3216), HeLa (CCL-2), 22Rv1 (CRL-2505), PC-3 (CRL-1435), DU-145 (HTB-81), LNCaP (CRL-1740), H1299 (CRL-5803), A-549 (CCL-185). Other cell lines: HVT (Science Cell, 7120), MOLM-13 (DSMZ, ACC554), COV-434 (Sigma Aldrich, 07071909). The SUM-149PT, MDA-MB-231, MCF-7, U87-MG, U-373, T-98G, LN-18, and A-172 cell lines were kind gifts from Tugba Bagcı Onder (Koc University, Istanbul, Turkey).

### 2.5. Cell Culture

Jurkat, K-562, MOLM-13, THP-1, 22Rv1, PC-3, DU-145, LNCaP were grown in RPMI-1640 (Sigma Aldrich #R8758) medium supplemented with heat-inactivated 10% fetal bovine serum (FBS; #1810 Biowest, Loire, France), 100 U/mL of penicillin, and 100 μg/mL of streptomycin (Life Technologies, Gaithersburg, MD, USA) at 37 °C and 5% CO2. The HEK293-T, HeLa, COV-434, SUM149PT, MDA-MB-231, MCF-7, U87-MG, U373-MG, T-98G, LN-18, and A-172 cells were grown in DMEM High Glucose (Sigma Aldrich #D6429) medium supplemented with heat-inactivated 10% fetal bovine serum (FBS; #1810 Biowest Loire, France), 100 U/mL of penicillin, and 100 μg/mL of streptomycin (Life Technologies, Gaithersburg, MD, USA) at 37 °C and 5% CO_2_. A-549, HVT, and H1299 cells were grown in DMEM-F12 (Sigma Aldrich #D6421) medium supplemented with heat-inactivated 10% fetal bovine serum (FBS; Biowest #1810), 100 U/mL of penicillin, and 100 μg/mL of streptomycin (Life Technologies) at 37 °C and 5% CO_2_. Cells were routinely tested for mycoplasma contamination using Lonza Myco Alert Detection Kit, (#LT07-318, Lonza Rockland, Rockland, ME, USA).

### 2.6. Cellular Thermal Shift Assay

THP-1 cells, 1 × 10^7^ cells at a density of 1 × 10^6^ cells/mL, were treated with 4 μM of test compounds (1 h, 37 °C, 5% CO_2_). Cells were centrifuged, and the pellet was resuspended with 1 × Phosphate Buffer Saline (Gibco, Grand Island, NY, USA, #10010023) containing 1 × Protease Inhibitor Cocktail (Cell Signaling Technology, Beverly, MA, USA, #5871) and 1 mM phenylmethylsulfonyl fluoride (PMSF, Sigma Aldrich, St. Louis, MO, USA, #10837091001). The suspension volume was divided equally and heated for 3 min at 43–64 °C in a T100 Thermal Cycler (Bio-rad, Hercules, CA, USA). For protein extraction, cells were snap-frozen using liquid nitrogen, thawed twice (25 °C, 1 min), then centrifuged (20,000× *g*, 15 min, 4 °C). Following centrifugation, protein containing supernatants were denatured at 70 °C for 10 min and loaded on SDS-PAGE gels (Bio-Rad, Hercules, CA, USA, 161-0317). After electrophoresis (100 V and 60 min), proteins were transferred onto PVDF membrane (Bio-Rad #1620177) at 200 mA for 180 min and immunoblotted with primary antibodies (1:10,000 dilution for anti-LSD1 Antibody ab129195 [EPR6825], 1:1000 dilution for Anti-HDAC6 Antibody CST 7558, 1:1000 dilution for Anti-HDAC1 Antibody ab53091, 1:1000 dilution for Anti-HDAC8 Antibody ab187139, 1:5000 dilution for Anti-beta Actin Antibody ab8227) overnight at 4 °C. The next day, membranes were incubated with HRP-conjugated secondary antibodies (1:5000 dilution for Abcam, Cambridge, UK, ab205718) for 1 h at room temperature. The signal was detected with BIORAD Chemi-Doc XRS+ System and by using Immobilon Forte Western HRP Substrate (WBLUF0500).

### 2.7. Cell Viability Assay

Cell viability of suspension cells was measured with Promega Cell Titer-Glo^®^ (Promega, Madison, WI, USA) Luminescent Cell Viability Assay. For each cell line, 5 × 10^3^ cells were seeded on 96 well plates. Following drug treatment for 120 h, cell titer-glo substrate (G755A) and cell titer-glo buffer (G756A) were mixed in 1:1 ratio. This mixture was further diluted with cell suspension in 1:10 ratio in 96 well plates (20 µL/well) that were incubated at 37 °C for 20–30 min. The luminescence signal was detected with a Biotek H1 Synergy microplate reader, and cell viability was normalized to vehicle treated cells. Each experiment was performed in duplicates with at least three biological repeats. For the combination experiments with GSK2879552 and ricolinostat, MOLM-13 cells (4 × 10^3^ cells/well) were co-treated with GSK2879552 and ricolinostat in all possible combinations of three different doses (0.6, 1.25, 2.5 μM) for 72 h. Cell viability was measured using ATP-dependent CTG assay as described above.

### 2.8. Protein Extraction

Cells were centrifuged (300 g, 5 min), and pellets were resuspended with RIPA Lysis Buffer (Sigma Aldrich, #R0278) containing 1mM PMSF (Sigma Aldrich, St. Louis, MO, USA, #10837091001) and 1 × Protease Inhibitor Cocktail (Cell Signaling Technology, Beverly, MA, USA, #5871) (10^6^ cells/mL). The suspension was centrifuged at 17,000× *g*, 20 min at 4 °C. For histone extraction, cells were collected (300 g, 5 min), and the pellet was resuspended with Triton Extraction Buffer containing 2 mM PMSF (Sigma Aldrich, #10837091001) at a final concentration of 1 × 10^6^ cells/mL and incubated on ice for 10 min. The cell suspension was lysed and centrifuged twice at 6500× *g* for 10 min at 4 °C. The supernatant was discarded, and the pellet was resuspended with 0.2N HCl at a final concentration of 4 × 10^6^ cells/mL overnight at 4 °C. The next day, samples were centrifuged again, and the supernatant containing histone proteins was normalized with 1:4 volume of 0.1 M NaOH.

### 2.9. SDS-PAGE Gel Electrophoresis and Western Blotting

Protein concentrations were detected with Pierce™ BCA Protein Assay Kit (Thermo Scientific™, Waltham, MA, USA, 23225). Samples (15–30 µg of whole protein and 2 µg of histone lysate) were loaded on 4–15% SDS-PAGE gel (Mini-PROTEAN TGX™ Precast Gels, Bio-rad, Hercules, CA, USA, #161-0317). Following electrophoresis (100 V and 60 min), separated proteins were transferred onto the PVDF membrane (Bio-Rad, Hercules, CA, USA, #1620177) at 200 mA for 150 min and immunoblotted with primary antibodies (1:1.000 dilution for Cell Signaling Technology (CST, Beverly, MA, USA) Acetyl-α-Tubulin (Lys40) (D20G3) XP^®^ Rabbit mAb 5335, 1:1000 dilution for Abcam (Abcam, Cambridge, UK) Anti-α-Tubulin antibody ab18251, 1:1000 dilution for Abcam Anti-Histone H3 antibody ab18521, 1:1000 dilution for Abcam Anti-Histone H3 dimethyl K4 antibody ab7766, 1:1000 for Cell Signaling Technology Acetyl-Histone H3 (Lys18) Antibody #9675, 1:1000 dilution for Cell Signaling Technology Acetyl-Histone H3 (Lys27) (D5E4) XP^®^ Rabbit mAb #8173, 1:10,000 diluted Anti-LSD1 Antibody ab129195 [EPR6825]) overnight at 4 °C. The next day, the membrane was incubated with HRP-conjugated secondary antibodies (1:5000 dilution for Abcam, ab205718) for 1 h at room temperature. The signal was detected with BIORAD Chemi-Doc XRS+ System. Band intensities were quantified using ImageJ software (NIH, Bethesda, MD, USA).

### 2.10. Combination Treatments with Antileukemic Drugs

Experiments with THP-1 cells (4 × 10^4^ cells/mL) were initiated by treatment with the dual inhibitor (5 μM) for 72 h. This was followed by a further treatment with iDual (5 μM) and a sublethal concentration of one of the following antileukemic drugs: cytarabine (0.62 μM), cisplatin (1.25 μM), paclitaxel (3 nM), or doxorubicin (62.5 nM), and the cells were incubated for 48 h. Similar experiments were performed with GSK2879552 and iHDAC6 (5 μM of each) or ricolinostat (1.25 μM) in combination with doxorubicin and the combination of iDual (5 μM) and etoposide (0.2 μM).

With the MOLM-13 cell line, 4 × 10^4^ cells/mL were treated with the iDual (0.5 μM) for 72 h. This was followed by a further treatment with iDual (0.5 μM) and a sublethal concentration of doxorubicin (either 2 or 4 nM), and the cells were incubated for 48 h. Similar experiments were performed with GSK2879552 (1.25 μM), iHDAC6 (0.5 μM) or ricolinostat (0.25 μM) in combination with doxorubicin (either 2 or 4 nM).

### 2.11. DNA Damage Repair/ABC/Sensitizer Gene Expression

THP-1 cells (2 × 10^5^) were treated with either iDual, iHDAC6, or GSK2879552 (5 μM in each case) or ricolinostat (1.25 μM) for 72 h and compared to vehicle control (DMSO). NucleoSpin™ RNA Plus Isolation Kit (Macherey Nagel™, Dueren, Germany, 740955.50) and M-MLV Reverse Transcriptase kit (Thermo Fisher Scientific, Waltham, MA, USA, #28025013) protocols were employed for RNA isolation and cDNA synthesis, respectively, with 500 ng of RNA converted to cDNA for each sample. Gene expression levels were quantified with qRT–PCR method using a LightCycler 480 SYBR Green I Master (Roche Diagnostics, Penzberg, Germany, #50-720-3180) and normalized to β-actin or GAPDH. Briefly, 10 ng of cDNA template was amplified using 10 μL LightCycler 480 SYBR Green I Master (Roche Diagnostics), and gene-specific primers (Appendix A) mixed in a final volume of 20 μL. The samples were incubated at 95 °C for 5 minutes, followed by 40 cycles of 95 °C for 10 seconds, 55 °C for 30 seconds, and 72 °C for 1 second. The relative amounts of gene expressions were calculated with the 2^(−ΔΔCT)^ method. All analyses were performed in duplicates and three biological replicas.

### 2.12. Mechanistic Studies with Doxorubicin Combinations

THP-1 cells (2 × 10^5^) were treated with either iDual, iHDAC6, or GSK2879552 (5 μM in each case) or ricolinostat (1.25 μM) for 72 h. This was followed by a further treatment with the compound (5 μM, or 1.25 μM for ricolinostat) and doxorubicin (62.5 nM) for the following time periods: 6 h for monitoring γH2Ax activation; 24 h for determining ROS elevation; 48 h for measuring apoptosis levels.

### 2.13. Annexin V Staining

Annexin V positivity was evaluated, as previously described, with minor changes [16]. Treated cells were centrifuged at 300× *g* for 5 min, and the pellet was resuspended with 1% FBS in PBS (5 × 10^5^ cells/mL). Annexin V dye (Merck Millipore, Burlington, MA, USA, MCH100105) was mixed with the cell suspension in a 1:1 ratio and the mixture incubated at RT for 20 min. Early/late apoptotic cells were counted with Muse Cell Analyzer (Merck Millipore, Burlington, MA, USA).

### 2.14. Caspase 3/7 Staining

Caspase 3/7 activation was evaluated, as previously described, with minor changes [17]. 150 μL of a working solution was prepared with Caspase 3/7 reagent (Merck Millipore, Burlington, MA, USA, MCH4700-1505) mixed with PBS in 1:8 ratio. The working solution was mixed with 50 µL of cell suspension and incubated for 30 min at 37 °C. Following that, 7-AAD dye (MCH4700-1505) was diluted with Buffer BA MCH4700-1360 (1:75 ratio). 7-AAD solution, 150 µL, was mixed with THP-1 cells (5 × 10^4^), and the suspension was incubated for 5 min at RT. Early/late apoptotic cells were counted with Muse Cell Analyzer (Merck Millipore).

### 2.15. γH2Ax Activation Determination

The Muse ɣH2A.X Activation Dual Detection Kit (Merck Millipore, Burlington, MA, USA, MCH200101) protocols were followed for double strand break analysis. Treated cells were centrifuged at 300× *g* for 5 min, and the pellet was resuspended with 1 × Assay Buffer (CS202134) and fixed on ice for 5 min with Fixation Buffer (CS202122) in 1:1 ratio. Fixed cells were centrifuged for 600 g for 5 min and mixed with Permeabilization Buffer (1:2 ratio CS203284) and incubated on ice for 5 min. Then, cells were centrifuged at 600 g and resuspended with 20 × Anti-H2A.X PECy5 (CS208202) and 20 × Anti-phosphate (Ser139) Alexa Fluor 555 (CS208203) antibody mix and incubated for 30 min at room temperature. Stained cells were resuspended with 200 μL of 1 × Assay Buffer and γH2AX activation was analyzed with Muse Cell Analyzer (Merck Millipore).

### 2.16. Reactive Oxygen Species Activation

Activation was assessed by Muse Oxidative Stress Kit (Merck Millipore, Burlington, MA, USA, MCH100111) with a similar protocol, which was previously described in [18]. Briefly, treated cells were centrifuged (300× *g*, 5 min), resuspended with 1 × Assay Buffer (MCH100111-2), and mixed with Oxidative Stress Reagent (4700-1665) in 1:100 ratio. A working solution was prepared by further dilution of 1 × Assay Buffer in 1:80 ratio. The cell suspension, 50 μL, was mixed with 150 μL working solution and incubated at 37 °C for 30 min. The data was analyzed with Muse Cell Analyzer (Merck Millipore).

## 3. Results and Discussion

### 3.1. IDual inhibits LSD1 and HDAC6

GSK2879552 is a clinical candidate that is an analogue of tranylcypromine (indicated in blue, Figure 1) with high selectivity for LSD1 and undetectable levels of inhibition against mechanistically related enzymes such as LSD2 or MAOs, and favorable in vivo oral bioavailability [19,20]. We anticipated that these characteristics would be retained in iDual, a closely related analogue in which the carboxylic acid is replaced by a hydroxamic acid (indicated in red, Figure 1). At the same time, incorporation of the hydroxamic acid should enable iDual to additionally target HDACs. In the design of dual mechanism inhibitors, it is important to have closely related control probes to ensure that the observed pharmacology is due to multitargeting rather than adventitious improvements in pharmacokinetics. For this reason, we prepared two additional compounds that are isomeric with iDual and GSK2879552, respectively, in which the altered position of the phenyl ring within the cyclopropyl ring leads to loss of activity against LSD1. In iHDAC6, this should result in HDAC inhibition only, whereas the null compound iNC should be inactive against either HDACs or LSD1.

We first profiled GSK2879552 and the novel compounds iDual, iHDAC6, and iNC in LSD1 and HDAC enzymatic assays. To gain an idea of HDAC selectivity, we tested for activity against the cytoplasmic isozyme HDAC6, the nuclear isozymes HDAC1 and HDAC3 that are strongly implicated in histone deacetylation, and the deacetylase/fatty acid deacylase HDAC8. As expected, the results (Table 1) indicated that GSK2879552 behaved as a positive control that inhibited LSD1 but not the HDAC isozymes. Interestingly, our dual inhibitor was approximately three-fold more potent against LSD1 than GSK2879552 despite their close structural similarity. In addition, it inhibited HDAC6 and HDAC8 with sub-micromolar IC_50_ values and displayed a pronounced selectivity for these two enzymes over the nuclear isozymes HDAC1 and HDAC3. Molecular docking against HDAC1/HDAC6 predicts a tighter bidentate coordination of iDual with HDAC6 (Appendix A), in support of the observed results. The isomeric cyclopropylamine iHDAC6 was inactive against LSD1 while being a respectable HDAC6/HDAC8 inhibitor, albeit lower in potency than iDual. This selectivity profile is often observed with aromatic hydroxamic acids such as iDual and iHDAC6. While others have reported dual LSD1/HDAC inhibitors, these compounds exhibited a different pattern of isozyme selectivity and inhibited nuclear HDACs that are strongly implicated in histone deacetylation [21,22,23]. We note that both iDual and iHDAC6 have >30-fold HDAC6/HDAC1 isoform selectivity, which is markedly higher than the approximately 10-fold level reported for the selective HDAC6 inhibitor ricolinostat in clinical trials [24]. As probes, GSK2879552 and iHDAC6 satisfied our requirements as single mechanism positive controls for comparison with the dual inhibitor. Finally, the negative control, iNC, was inactive against either LSD1 or HDACs.

Cellular thermal shift assay (CETSA), which relies on the principle of thermal stabilization of the target protein upon drug binding (Figure 2A), was used to assess the ligand engagement of the dual inhibitor. The CETSA results confirm that iDual increased thermal stability of both LSD1 and HDAC6, as more soluble protein was detected at higher temperatures compared to untreated samples (Figure 2B, compare 49–52 °C for LSD1 and 49–58 °C for HDAC6; quantifications of band intensity in Figure 2C). On the other hand, HDAC8 was not significantly stabilized, suggesting low cellular target engagement with this isozyme. Meanwhile, our positive controls behaved as predicted, with GSK2879552 stabilizing LSD1 only, and ricolinostat stabilizing HDAC6 (Appendix A). Interestingly, whilst iHDAC6 stabilized HDAC6 only, ricolinostat not only stabilized HDAC6, but also HDAC1 and to a lesser extent HDAC8, indicating that iDual and iHDAC6 are more selective than ricolinostat in cells, despite the higher IC_50_ values in biochemical assays (Table 2). As expected, the negative control, iNC, did not exhibit target engagement with either LSD1 or HDACs (Appendix A).

### 3.2. LSD1 Is Overexpressed in Leukemic Cell Lines

Overexpression of LSD1 is frequently observed in many cancer types, including hematopoietic malignancies [25,26]. Therefore, the expression of LSD1 was evaluated by Western blotting in a panel of 21 cancer cell lines. As reported earlier, its expression was found to be elevated in several of these lines (such as leukemia, breast, cervix, ovarian, prostate, glioma, and lung cancers) (Figure 3A) in comparison to the non-cancerous trophoblasts (HVT cells). Our data revealed that the highest LSD1 expression was in a leukemia cell line (THP-1), followed by 22Rv1 a prostate cancer cell line (Figure 3B).

In support of this observation, when we curated RNA-seq based data from patient derived samples [27], LSD1 expression was significantly elevated in many different tumors compared to normal tissue (Appendix A), with the highest fold (5.12) change in AML samples (Appendix A).

### 3.3. IDual Reduces Leukemic Cell Viability

Single mechanism of action inhibitors of either LSD1 or HDACs have progressed to clinical trials for the treatment of AML. Although neither enzyme is frequently mutated nor highly expressed in AML, they contribute to the heterogeneity characteristic of this cancer and promote malignancy through the silencing of myeloid differentiation and other antiproliferative cell fates [28,29]. For our cellular profiling, we selected a set of four leukemia cell lines that express LSD1 at high levels (Figure 3). Within this panel, K-562 and Jurkat cells carry the wild-type MLL gene, whereas THP-1 and MOLM-13 contain an oncogenic MLL-AF9 fusion [30]. The latter chromosomal translocation is known to be particularly sensitive to both LSD1 and HDAC inhibition [31,32].

The cells were treated with the inhibitors for 120 h in the concentration range of 0.6–20 μM for THP-1, K-562, and Jurkat and at the lower range of 0.1–20 μM for the more sensitive MOLM-13 cells. The half-maximal effective doses were calculated (Table 2) from the cell viability curves (Appendix A). The cell viability assay results (Table 2) indicate that iDual is more potent than GSK2879552 in the growth inhibition of all four leukemia cell lines, while it had no significant toxicity on normal RPE-1 cells (Appendix A). Gratifyingly, the negative control iNC was inactive at the tested concentrations, implying that the GSK2879552 scaffold is relatively selective in its cellular binding interactions and not subject to appreciable off-target cytotoxicity. Meanwhile, our HDAC6 positive control ricolinostat was the most potent compound across the board, and this probably arises through a combination of the lower IC_50_ against HDAC6 and the poorer HDAC6/nuclear HDAC isozyme selectivity relative to iDual and iHDAC6. While iHDAC6 had a similar profile to the dual inhibitor iDual, it was less effective against MOLM-13, suggesting that this cell line is more sensitive to the dual inhibition of LSD1 and HDAC6. Our results suggest a similar outcome could be achieved through the combination of single mechanism LSD1 and HDAC6 inhibitors. While our work was in progress, dual inhibition of LSD1 and HDAC6 was shown to be beneficial in cell-based and animal models of multiple myeloma as well [33].

### 3.4. IDual Increases Cellular Protein Methylation/Acetylation and Induces Cell Differentiation of THP-1 Cells

In support of cellular LSD1 inhibition, THP-1 cells treated with either GSK2879552 or iDual displayed an increase in the levels of the histone substrate H3K4me2 that were comparable to each other (Figure 4A). The dual inhibitor increased nuclear histone H3 acetylation two-fold, as monitored by H3K18ac and H3K27ac levels (Figure 4B). On the other hand, ricolinostat had a greater effect at similar half inhibitory doses with iDual (~0.75 μM for ricolinostat and ~4 μM for iDual; ~25% of IC50 values based on Table 2) were used, or when cells were treated at equal concentrations (4 μM), (Figure 4B, purple bars), which may be attributed to the lower isozyme selectivity of ricolinostat. The acetylation levels of α-tubulin, a cytoplasmic HDAC6 substrate, were robustly increased by approximately 10-fold in THP-1 cells dosed with either iDual and by 20-fold when similar IC_50_ of ricolinostat was used and 25-fold at 4 μM of ricolinostat (Figure 4C). As HDAC6 activity also affects cytoplasmic proteins that regulate cell motility or adhesion [34], we explored the functional consequence related to these targets after iDual treatment and performed wound healing assays with adherent DU-145 and MDA-MB-231 cells. As expected, HDAC6 inhibition slowed down the wound healing capacity of both cell lines albeit mildly (Appendix A).

Since LSD1 silencing or inhibition by itself is reported to increase histone acetylation [35,36], we checked if this might contribute to the effect observed with the dual inhibitor. However, the control GSK2879552, acting by LSD1 inhibition alone, did not significantly alter histone acetylation levels (Appendix A).

A hallmark of cellular HDAC inhibition is the increased expression of the cyclin-dependent kinase p21 [37,38,39,40]. This effect is not only observed with pan HDAC inhibition, as selective inhibition of HDAC6 has also been shown to increase p21 expression in leukemia [41] and glioblastoma cells [42]. In accordance with these results, both iDual and the control probe iHDAC6 increased p21 levels (Figure 4D), but to a lesser extent compared to ricolinostat, which once again might arise from its greater inhibition of nuclear histone deacetylation (Appendix A).

The Induction of CD86 and CD11b, two surface proteins associated with myeloid differentiation, is widely used as a downstream biomarker of LSD1 inhibition in leukemia cell lines [43]. In THP-1 treated cells, the dual inhibitor iDual produced a much higher fold change in the expression of both CD86 and CD11b than GSK2879552, indicating a positive benefit of the simultaneous targeting of LSD1 and HDAC6 (Figure 4E).

### 3.5. iDual Sensitizes AML Cells to Doxorubicin

Epigenetic reprogramming is an attractive option for the priming of tumor cells to respond to other treatments [44,45]. We investigated the potential of iDual in combination therapy with four established antileukemic drugs, each with a distinct mechanism of action: cytarabine, cisplatin, paclitaxel, and doxorubicin. THP-1 cells were incubated with iDual (5 μM) for 72 h, followed by co-treatment with a second dose of iDual and a sublethal concentration of the antileukemic drug, and viability was measured via ATP Cell Viability Luciferase Assay (Figure 5A). Interestingly, a significant decrease in cell viability was observed only with the combination of iDual and doxorubicin in both THP-1 (Figure 5B), in more than two doses (Appendix A), and MOLM-13 cells (Appendix A).

A similar response (Figure 5C) was also detected upon pre-treatment with the single mechanism agents GSK2879552 and iHDAC6 (both at 5 μM) and, to a much lesser extent, with ricolinostat (at a lower concentration of 1.25 μM due to its higher cytotoxicity). Meanwhile, decreased cell viability was not observed with the combination of the dual inhibitor and etoposide (Figure 5D), a topoisomerase II inhibitor similar to doxorubicin. While there are several reports of enhancement of doxorubicin cytotoxicity by a HDAC6 selective inhibitor, these however involved higher concentrations and simultaneous administration of the two agents without initial epigenetic priming [46,47]. As for LSD1 inhibition in combination with doxorubicin, there is one example of pre-treatment resulting in an increased antiproliferative effect in MCF-7 breast cancer cells [48].

We repeated the priming experiments with LSD1/HDAC inhibitors in the MOLM-13 cell line that also features the MLL-AF9 gene fusion using two sublethal concentrations of doxorubicin, 2 or 4 nM (Appendix A). At the lower dose of 2 nM doxorubicin, the greatest decrease in cell viability occurred when cells were primed with the iDual. Ricolinostat was next in terms of activity, while the positive controls GSK2879952 and iHDAC6 were relatively ineffective. With 4 nM of doxorubicin, a significant decrease was seen with all compounds except iHDAC6. Overall, our results demonstrate that priming by LSD1 or HDAC6 inhibition, and preferably both, can greatly enhance the sensitivity of leukemia cell lines with MLL translocations to a sublethal concentration of doxorubicin.

We probed the consequence of the epigenetic pre-treatment upon programmed cell death induced by doxorubicin. At a sublethal concentration, the dual inhibitor did not induce apoptosis, as determined by annexin V/7-AAD staining (Figure 6A and Appendix A) and caspase activity (Figure 6B and Appendix A). According to the caspase 3/7 staining, while doxorubicin alone at the sublethal concentration was only able to trigger mild apoptosis with 30% of cells in the apoptosis/dead quadrant, pre-treatment with iDual resulted in a dramatic response of over 80%. Furthermore, the dual inhibitor produced a more pronounced caspase 3/7 effect compared to the single mechanism agents GSK2879552, iHDAC6, or ricolinostat.

One of doxorubicin’s major mechanisms of cytotoxicity is through the inhibition of topoisomerase II, in addition to induction of double strand breaks and reactive oxygen species (Figure 7A) [49,50]. However, we believe this was not a significant contributor in our experiments, as sensitivity was not observed with etoposide, another topoisomerase II inhibitor (Figure 5C). Through their effect on topoisomerase II, both doxorubicin and etoposide produce double-strand DNA breaks (DSBs), resulting in the rapid accumulation of phosphorylation in the histone H2A variant γH2AX [51]. Accordingly, we observed an increase in the formation of DSBs (up to 40%) in THP-1 cells upon treatment with doxorubicin (Figure 7B and Appendix A). On the other hand, neither iDual, GSK2879552, nor ricolinostat resulted in a significant enhancement of DSB levels, either on their own or as a pretreatment to doxorubicin. Meanwhile, iHDAC6 led to a mild increase in DSBs compared to doxorubicin alone, consistent with a previous report using a selective HDAC6 inhibitor (Appendix A) [47]. In addition to topoisomerase II inhibition or DNA intercalation, doxorubicin has a DNA-independent mechanism of action that involves the generation of reactive oxygen species (ROS), which then trigger pathways of cell death. While we observed ROS generation in doxorubicin treated cells compared to controls, no further increase occurred in combination experiments of doxorubicin with the LSD1 inhibitor GSK2879552, HDAC inhibitors, iHDAC6, ricolinostat, or with the dual inhibitor (Figure 7C, Appendix A). Overall, our results highlight the unique features of sequential treatments with sublethal drug concentrations, thus avoiding the major pathways of doxorubicin cytotoxicity.

To assess the potential gene expression changes that may contribute to doxorubicin sensitivity, we initially monitored the levels of fifteen key DNA repair genes from five different mechanistic groups in THP-1 cells treated with the dual inhibitor. We hypothesized the resulting DNA damage would induce DNA repair mechanisms and sensitization by our epigenetic modulators would reduce the DNA damage response. Among the tested genes, only the homology-directed repair pathway genes showed an increase in expression levels and. In particular, Rad51 showed an almost three-fold increase (*p* ≤ 0.001) (Figure 7D), followed by ATM and ATR with a lower ~two-fold increase. This suggests that, while there may be a degree of regulation within this pathway, the set of genes studied herein did not undergo reduced expression. Next, we set to determine whether regulation of drug efflux pumps play a role in the observed doxorubicin sensitivity. Since ABC family transporters are mainly responsible for the efflux of doxorubicin [52], we measured their expression and found no decrease of a panel of ABC transporters (Figure 7E). On the contrary, iDual treatment significantly upregulated some of the efflux pumps, which indicates that other mechanisms, which surpass all these changes, occur in response to inhibition of HDAC6 and LSD1. Therefore, we hand-picked several genes that were reported to be involved in cellular response to doxorubicin (Table 3). All these targets, except for Wnt7B, were upregulated upon iDual treatment (Figure 7F), potentially explaining how the dual inhibitor sensitizes AML cells to doxorubicin. While a cumulative action of all these genes may account for the observed sensitization, some may contribute more than the others.

In summary, our results indicate that dual LSD1/HDAC6 inhibition successfully augments doxorubicin treatment in a cooperative manner to activate apoptosis. Notably, our sequential dosing achieved cell death with a normally sublethal concentration of both the epigenetic inhibitor and doxorubicin. Moreover, this apparently did not arise from the widely documented effects of doxorubicin, such as topoisomerase II inhibition, DNA intercalation, or the generation of ROS. Further experiments are underway to identify the underlying mechanisms, with chromatin damage being one possibility noted in a recent study [60].

## 4. Conclusions

We show that minimal structural perturbation of carboxylic acid to hydroxamic acid within the clinical candidate GSK2879552 produced a dual LSD1/HDAC6 inhibitor. We compared the pharmacology of iDual against GSK2879552 and iHDAC6, positive controls that act by a single mechanism, and the null compound iNC as a negative control. The dual inhibitor inhibited LSD1, HDAC6, and HDAC8 with sub-micromolar IC_50_ values. In fact, iDual gained three-fold potency relative to the candidate GSK2879552 against its LSD1 target and was more effective than the latter in the growth inhibition of two AML cell lines. The measurement of enzyme substrate levels, biomarkers, and CETSA confirmed that iDual exhibits dual target engagement in cells. In combination experiments, iDual, as well as the single mechanism agents iHDAC6 and GSK2879552, displayed significant sensitization of THP-1 and MOLM-13 cells, with MLL-AF9 gene fusions, towards the induction of apoptosis by a sublethal concentration of the clinical drug doxorubicin. The underlying mechanism for the increased sensitivity is not correlated with the major modes of action of doxorubicin encompassing topoisomerase II inhibition, increased DNA breaks, or ROS elevation. We observed that iDual increased expression of genes previously correlated with doxorubicin sensitivity and speculate that a cumulative mechanism involving these genes is responsible for the synergy. Our results indicate that priming by inhibitors targeting LSD1 and HDAC6 is a promising approach to reduce the therapeutic dosing of doxorubicin and thereby address adverse events, such as the cardiotoxicity that is currently a major limitation for its clinical application.

## Figures and Tables

**Figure 1 cancers-14-06014-f001:**
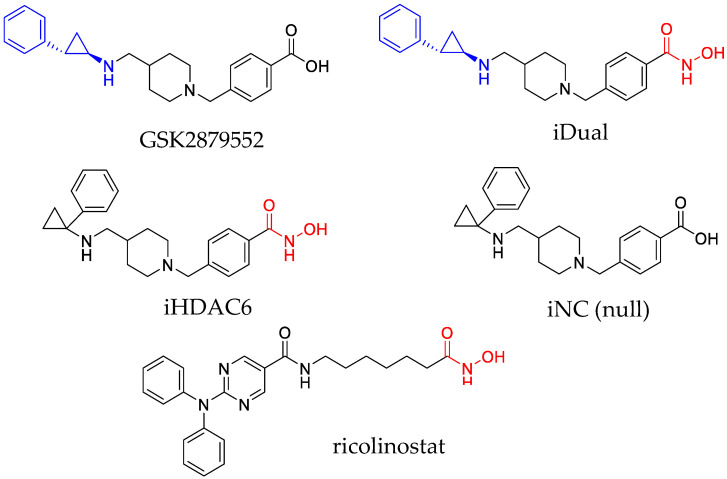
Structures of clinical candidates GSK2879552 and ricolinostat (targeting LSD1 and HDAC6, respectively), and the new analogues: dual inhibitor iDual, HDAC6 inhibitor (iHDAC6), and the negative control (iNC). The tranylcypromine and hydroxamic acid pharmacophores required for LSD1 and HDAC inhibition are highlighted, respectively, in blue and red.

**Figure 2 cancers-14-06014-f002:**
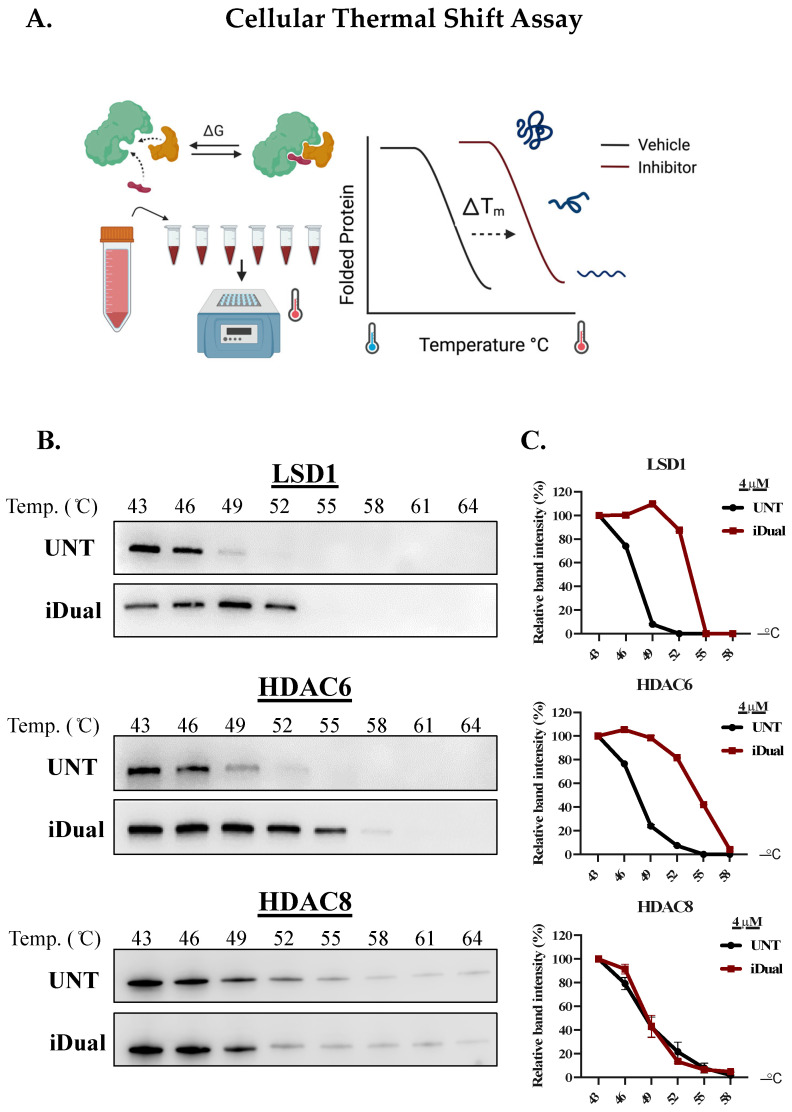
iDual is a dual inhibitor targeting LSD1, HDAC6, and HDAC8. (**A**) Cellular thermal shift assay (CETSA) experimental procedure was generated in BioRender.com. (**B**) Drug and target engagement and protein stabilization results via CETSA. LSD1 and HDAC6/8 enzymes in THP-1 cells treated with DMSO (untreated, UNT) or the dual inhibitor iDual (4 μM, 1 h). Full Western blot images can be found in Appendix A. (**C**) Relative band intensities quantified using Image J software and indicated graphically for LSD1, HDAC6, and HDAC8.

**Figure 3 cancers-14-06014-f003:**
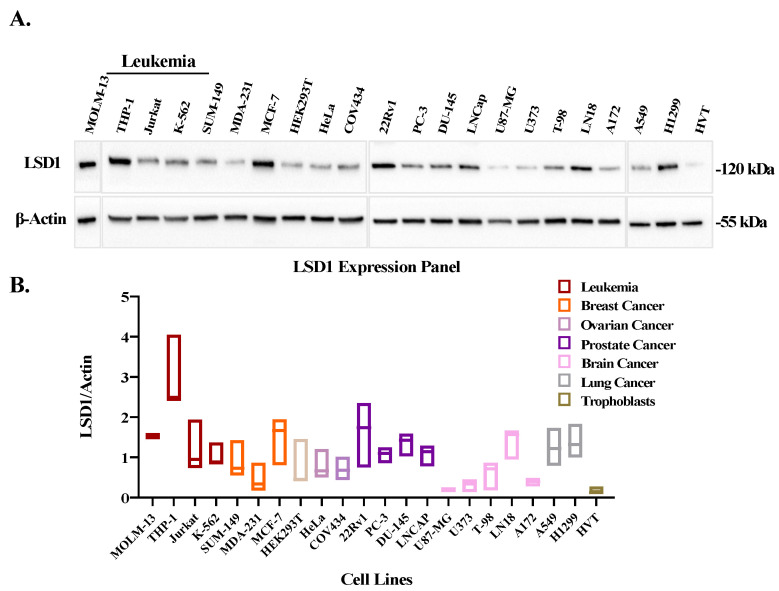
LSD1 expression profile of different cancer cell lines and non-cancerous HVT cells. (**A**) Western blot analysis of the LSD1 expression in different cancer types. Full Western blot images can be found in Appendix A. (**B**) quantification of LSD1 levels normalized to actin using Image J software.

**Figure 4 cancers-14-06014-f004:**
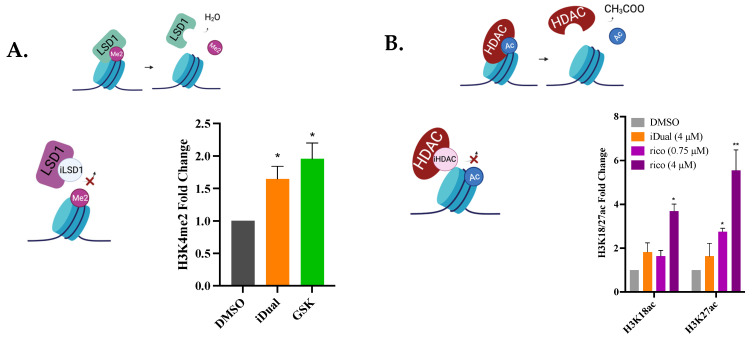
Assessment of biological activity through changes in methylation or acetylation and cell differentiation (**A**–**C**). Colored graphs illustrate epidrug targets and visualize expected outcomes upon target inhibition. Blue cylinders represent histone octomers (**A**,**B**) and green tubes represent microtubules (**C**). Fold change in methylation and acetylation levels of LSD1 and HDAC substrates between THP-1 cells treated with DMSO (control) or inhibitors iDual, iHDAC6, GSK2879552 (4 μM, 24 h), or ricolinostat (0.75 μM or 4 μM, 24 h). Histone methylation, histone acetylation, and α-tubulin acetylation levels were quantified from Western blot band intensities (Appendix A). Full Western blot images can be found in Appendix A. (**D**,**E**) Expression levels of mRNA were measured by RT-qPCR analysis and were normalized to two housekeeping genes, β-actin and GAPDH, for p21 and CD11b and CD86. THP-1 cells were treated with DMSO (control) or compounds (1 μM, 24 h). Relative expression was determined with the 2(^−ΔΔCT^) method, and the data are represented as the average ± standard error of the mean of two biological repeats with technical duplicates. (* *p* < 0.05, ** *p* < 0.01 student *t*-test). Illustrations were generated in BioRender.com.

**Figure 5 cancers-14-06014-f005:**
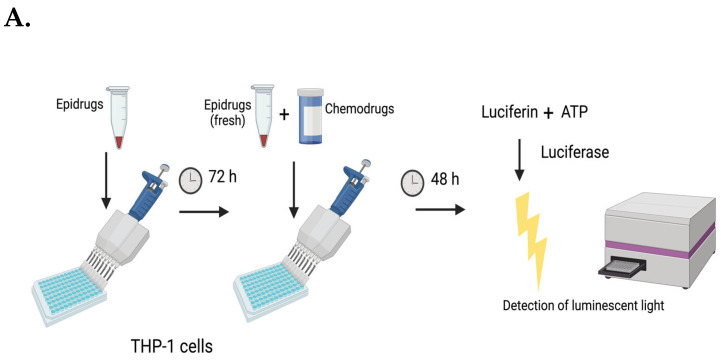
Cell viability in THP-1 cells after combination therapy with epigenetic inhibitors and clinically used agents. (**A**) Design of the THP-1 cell viability assays. Cells were pretreated with the epidrugs for 72 h and further incubated with the chemodrugs in combination with fresh epidrugs for an additional 48 h before viability was measured. Cell viability was measured using ATP-dependent cell-titer glo assay and normalized to control cells. The figure was generated in BioRender.com. (**B**) Pre-treatment with dual inhibitor 2 (iDual, 5 μM, 72 h) followed by co-treatment (48 h) with iDual (5 μM) and a sublethal concentration of the antileukemic drugs cytarabine (CYT, 0.62 μM), cisplatin (CISP, 1.25 μM), paclitaxel (PTX, 3 nM), or doxorubicin (DOX, 62.5 nM). (**C**) Pretreatment with iHDAC6, GSK2879552, or ricolinostat, followed by co-treatment with a sublethal concentration of doxorubicin. (**D**) Pre-treatment with dual inhibitor followed by co-treatment with a sublethal concentration of the topoisomerase II inhibitor etoposide (ETO, 0.2 μM). Data were represented as the average of ± standard deviation from three biological repeats (* *p* < 0.05, ** *p* < 0.01, *** *p* < 0.001 and n.s: not significant) ordinary one-way ANOVA, Tukey’s multiple comparisons test).

**Figure 6 cancers-14-06014-f006:**
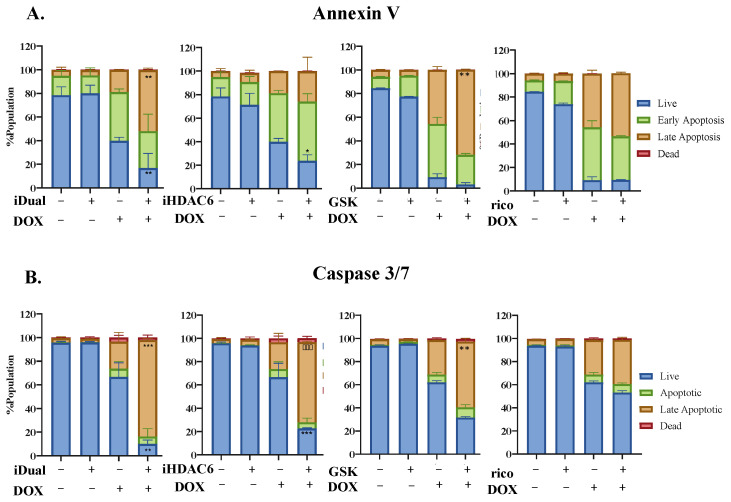
Flow cytometric analysis of doxorubicin combination in THP-1 cells. Apoptosis levels of THP-1 cells pretreated with iDual, iHDAC6 or GSK2879552 (5 μM, 72 h) or ricolinostat (1.25 μM, 72 h) followed by co-treatment with doxorubicin (62.5 nM, 48 h). (**A**) Annexin V or (**B**) caspase 3/7 and 7-AAD positive cells were analyzed by flow cytometry (Appendix A). Data were represented as the average of ± standard deviation from two biological repeats (* *p* < 0.05, ** *p* < 0.01 and *** *p* < 0.001, two-way ANOVA).

**Figure 7 cancers-14-06014-f007:**
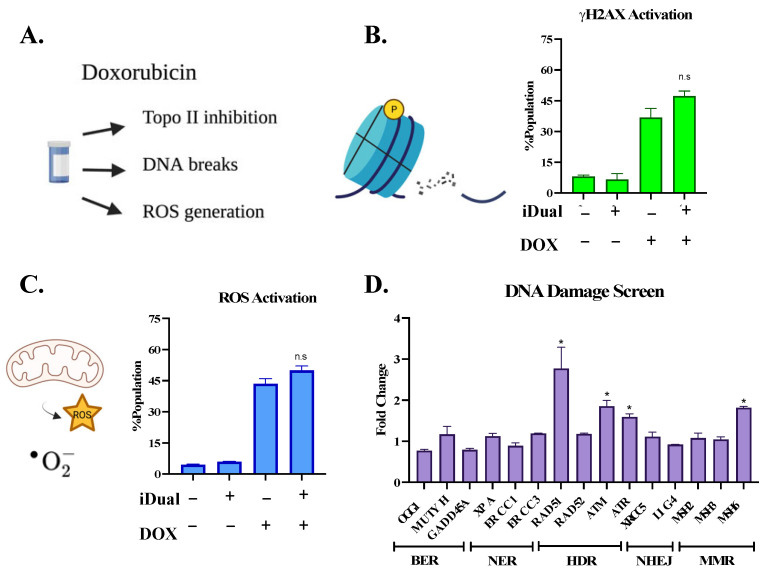
Investigation of doxorubicin synergy. (**A**) The mechanism of action for doxorubicin induced cytotoxicity is mainly through inhibition of topoisomerase II or induction of double strand breaks or reactive oxygen species (ROS). (**B**) Upon DSB formation, an H2B variant, H2AX, gets phosphorylated (γH2AX, yellow circle), which is a hallmark of DSBs. (The blue cyclinder represents the histone octomer.) Quantification of DNA DSBs as monitored by flow cytometry of γH2AX activation upon treatment of THP-1 cells with iDual (5 μM, 72 h) and doxorubicin (DOX, 62.5 nM, 48 h). (**C**) ROS levels of THP-1 cells with iDual, (5 μM, 72 h) and doxorubicin (62.5 nM, 24 h). Generation of oxidative stress was assessed by flow cytometry. (**D**) Expression levels of DNA damage repair genes from five pathways (BER: base excision repair; NER, nucleotide excision repair; HDR, homology directed repair; NHEJ, non-homologous ends joining; MMR, mismatch repair) upon treatment of THP-1 cells with iDual (5 μM, 72 h). (**E**) Doxorubicin is a substrate of ABC transporters (ATP pump, represented in pink), which can efficiently efflux drugs outside the cell, reducing their toxicity. RT-qPCR analysis showing expression levels of different ABC genes upon iDual (5 μM, 72 h) in THP-1 cells. Expression levels of sensitizer genes after iDual (5 μM, 72 h) via qPCR analysis in THP-1 cells (**F**). Relative RTq-PCR target gene expressions were determined by the 2^(−ΔΔCT)^ method. Data were represented as the average of ± standard error of the mean of two biological repeats with duplicates. (* *p* < 0.05, ** *p* < 0.01 and *** *p* < 0.001, student’s *t*-test). Schematic illustrations were generated in BioRender.com.

**Table 1 cancers-14-06014-t001:** LSD1, HDAC1, HDAC3, HDAC6, and HDAC8 enzyme assays with iDual, single mechanism positive controls GSK2879552 and iHDAC6, and the null compound iNC. IC_50_ values are ± standard error from three independent measurements.

IC_50_ (μM)					
Compound	LSD1	HDAC1	HDAC3	HDAC6	HDAC8
GSK2879552	1.54 ± 0.41	>100	>100	>100	>100
iDual	0.54 ± 0.03	5.31 ± 0.25	2.29 ± 0.14	0.11 ± 0.01	0.29 ± 0.04
iHDAC6	>100	7.57 ± 0.29	4.09 ± 0.26	0.23 ± 0.02	0.45 ± 0.07
iNC	>100	>100	>100	>100	>100

**Table 2 cancers-14-06014-t002:** Leukemia cell viability measured by the ATP-dependent cell titer-glo assay. IC_50_ values are normalized relative to control cells ± standard deviation from three biological repeats.

IC_50_ (μM)				
Compound	THP-1	MOLM-13	K-562	Jurkat
GSK2879552	>20	15.0 ± 3.5	>20	>20
ricolinostat	2.9 ± 0.1	0.2 ± 0.03	2.6 ± 0.3	1.9 ± 0.003
iDual	16.3 ± 5.4	1.3 ± 0.4	16.6 ± 2.7	15.5 ± 1.4
iHDAC6	9.7 ± 2.4	6.7 ± 1.6	17.6 ± 1.6	10.7 ± 0.8
iNC	>20	>20	>20	>20

**Table 3 cancers-14-06014-t003:** List of genes reported to exhibit synergy with doxorubicin in either AML or other cancer cell lines.

Name	Description	Ref.
AGPAT9	Increased AGPAT9 expression in MCF-7 cells increases chemosensitivity to doxorubicin	[53]
ALOX5	ALOX5 exhibits antitumor and doxorubicin-sensitizing effects in MLL-rearranged leukemia	[54]
BTG1	Doxorubicin-induced cell death is mediated by BTG family	[55]
HIPK2	HIPK2 regulates chemosensitivity including doxorubicin	[56]
IFI44L	Overexpression of IFI44L decreased chemoresistance towards doxorubicin	[57]
LRP1	Increased LRP1 may be associated with higher endocytosis of upregulated transporter proteins at the cell surface, thus increased doxorubicin accumulation	[58]
WNT7B	Wnt7B inhibits osteosarcoma cell growth, migration, invasion, and sensitivity to doxorubicin	[59]

## Data Availability

All chemistry and biological data are available on request. The agreement numbers for Confirmation of Publication and Licensing Rights for the images created in Biorender.com are: YZ24MZ62HY, AL24MZ7RA3, IU24MZ8A6I, DG24MZK162, KO24N9EINT, GW24MZ5H0L.

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
