# Peer review of "Dual LSD1 and HDAC6 Inhibition Induces Doxorubicin Sensitivity in Acute Myeloid Leukemia Cells"

_cancers, 2022, doi:10.3390/cancers14236014_

Round 1

Reviewer 1 Report

This manuscript described the development and biological evaluation of a dual LSD1/HDAC6 multi-targeting inhibitor (iDual). The dual targeting agent was superior to LSD1 inhibitor GSK2879552 in the growth inhibition of a panel of four AML cell lines. In combination experiments, the dual inhibitor primed AML cells to apoptosis with a sublethal concentration of doxorubicin. The study is interesting and has potential but requires a number of questions to be answered.

1. Chemical structures shown in Figure S4, S5, and S6 are incorrect.

2. The authors should show the evidence for the change of H3K4me2 and H3K18/27ac (Figure 4A and 4B) after treatment of iDual and GSK in the manuscript or the supporting information, such as western blot membrane.

3. In line 388, the authors state p21 is a biomarker for HDAC inhibition, they should cite reference for this. Is p21 is a biomarker for HDAC6 inhibition?

4. Are there any references or preliminary experiments to support the concentrations of antileukemic drugs cytarabine, cisplatin, paclitaxel and doxorubicin (Line413-414)? As only one concentration for each antileukemic drug was used, it is not adequate to conclude if iDual has synergistic effect with these drugs. And what is the treatment time for single cytarabine, cisplatin, paclitaxel and doxorubicin groups (Figure 5)?

5. The abstract is too long in the current version, the authors should reduce it to appropriate length.

Reviewer 2 Report

In this manuscript, Bulut et al describe the synthesis of a molecule (named iDual) with the ability to inhibit LSD1 and HDAC6. After having validated its efficacy, tests conducted on 21 cell lines show the increased efficacy of iDual on cell lines of leukemic origin, especially those with a translocation of MLL. Finally, the authors demonstrated that the apoptotic effect of doxorubicin was increased in combination with iDual, an effect independent of DNA repair mechanisms or efflux pumps.

This work is well written and has a lot of data.

Nevertheless, I have a few comments on this work:
Major remarks:
This work is performed on cell lines and only the THP-1 cell line is used to support the majority of the conclusions. In order for this work to have a more global scope on AML, each experiment should be performed on several cell lines and validated on patient cells.
Many experiments were performed with two replicates, which is insufficient. The most striking example is the lack of reproducibility in Figures 5 and 6 of the doxorubucin alone or iDual alone conditions. In particular, the iDual condition alone in the "etoposide" experiment leads to 115% viability, compared to 80% in the other experiments.

Minor remarks:
-supplemental figures should appear in the text in order (do not start with S13).
-Tables should show units of measurement.
-the text of the results should describe the results and not just show the conclusions of the experiments. For example, Figures 2 and 3 are neither detailed nor explained in the text, even though they include parts A, B and even C, which are never mentioned.
-The authors should clarify the terms "proliferation" and "viability", used alternately in the text.
-From Figure 4 onwards, the legends are poorly integrated into the text.
-the differentiation of cell lines must be validated by other techniques (cytology, flow cytometry...).
-Why did the authors choose a 5+3 treatment scheme (5 days of epidrugs + 3 days of chemodrugs)? have other treatment schemes been tested?
-line 446-448: how do the authors arrive at this conclusion. This needs to be detailed.
-ANOVA tests compare groups of data with a Gaussian distribution. I am not sure if this is applicable to this work. Also, if a post-test comparing each group was used, this should be specified. Same remark for the t-test.
-The legends of Figure 7C and 7D need to be revised as they do not correspond to what is shown.

Reviewer 3 Report

Relevant points:

-The AML cell lines tested at the beginning were only 2, not 4 as stated by the authors (Jurkat is not an AML cell line, but a T cell leukemia; K562 is a CML), weakening the importance of HDAC6+LSD1 inhibition in AML. Both the 2 AML cell lines harbor the MLL-AF9 rearrangement, restricting the relevance of the results on this rearrangements. However, there is not a not-MLL translocated counterpart to investigate in effects are related to the MLL rearrangement.

-In one of the 2 AML cell lines, the THP1, the iDUAL drug did not present advantages compared to iHDAC6 inhibitor or ricolinostat, both in cell viability as single agent (table 2 and Figure S15), and in cell viability and apoptosis induction in combo with DOX (Figure 5C and Figure 6). Again, these results lowered the relevance of a dual targeting in leukemia.

-A negative control as cells from a healthy bone marrow or PBMCs should be included to investigate toxicity of drug treatment.

-HDAC6 targets alpha-tubulin and HSP90 in the cytoplasm, regulating cell motility, adhesion and chaperone function (Zhang et al, Frontiers in Oncology 2021). The authors never explored the functional consequence related to these targets after iDual treatment.

Minor points:

-Figure S10-S11 and S12 are not cited along the manuscript.

-In figure S14, in the CETSA experiment, it seems that at 500 nM the iHDAC6 targets HDAC6 much more that iDual, that showed a profile which overlap with GSK2879552.

-Figure 3: it would be useful to compare LSD1 expression of leukemic cells with the normal counterpart, such as bone marrow cells of healthy donors or mobilized PBMCs.

-Line 344: the concentration used are reported to be 0.6-20 uM; however, in Table 2, IC50 of ricolinostat in MOLM-13 is 0.2 uM.

-Table 2: the unit of measurement is not specified in the caption.

-Line 358: in Figure S15A-D is not reported the co-administration of GSK2879552 and ricolinostat, as stated in the text.

-Line 340: The ref 29 reported that HDAC class I inhibitors impact on MLL-rearranged AML, but HDAC6 is a class IIb enzyme.

-Figure 4CB cited after 4C.

Reviewer 4 Report

Dual LSD1 and HDAC6 inhibition induces doxorubicin sensitivity in acute myeloid leukemia cells

The paper “Dual LSD1 and HDAC6 inhibition….” By Bulut et al. describes the development of a dual inhibitor (termed iDual) that is able to inhibit both LSD1 and HDAC6/8. The authors show that this dual inhibitor is able to better inhibit the growth of four different leukemic cell lines than an inhibitor of LSD1 that is under clinical investigation (GSK2879552). Further, they demonstrate that pre-treatment of the leukemic cell line THP-1 with iDual sensitizes the cells to doxorubicin induced killing using a sublethal concentration of doxorubicin, and suggest that inhibition of LSD1 and HDAC6 provide a means by which the toxicities associated with doxorubicin may be reduced. Whilst they have thus far failed to determine the precise mechanism of this sensitisation, they do show an increase in mRNA levels of a number of genes that have been reported to be involved in the cellular response to doxorubicin. However, no functional studies related to the increased expression of these genes were carried out.

Specific Comments:

1.      Page 2, line 89. What is the source of the GSK2879552 used in subsequent procedures?

2.      Page 3, line 135. What is the source of the cell lines used obtained, and where they regularly tested for mycoplasma contamination?

3.      Page 3, line 155. In the cellular thermal shift assays, the secondary antibody listed (ab205718 ) is an anti-rabbit antibody, yet the anti LSD-1 antibody used in the procedure (ab190507 ) is a goat antibody.

4.      Page 3, line 155. Which ECL reagent was used?

5.      Page 4, line 180. How was the protein concentration determined?

6.      Page 6, line 290. The authors comment that “both iDual and iHDAC6 have >30-fold HDAC6/HDAC1 isoform selectivity which is markedly higher than the approximately 5-fold level reported for the candidate ricolinostat”. Did the authors examine ricolinostat in their system?

7.      Page 7, line 302. In the table description, indicate that the IC50 values are in nM.

8.      Page 7, Table 1. As trichostatin was included as a positive control in the HDAC enzyme assays, include these values in the table.

9.      Page 7, Table 1. The authors have included the IC50 value for HDAC8. Can the authors comment on why HDAC8 inhibition was not included in subsequent experiments?

10.  Page 7, line 306. The authors state that the CETSA data confirms that iDual thermally stabilises HDAC6. In Figure S14 (both the western blots and the graphs), all of the compounds appear to show some degree of stabilisation of HDAC6, including the negative control compound (iNC). How many replicates were performed for this experiment, and was the effect of iDual found to be statistically significant?

11.  Page 7, line 307. The authors state that ricolinostat only stabilised HDAC6. This statement needs clarification, as it can be seen in Fig S14 that ricolinostat also stabilised HDAC1.

12.  Page 8, line 323. “Our data revealed that the highest LSD1 expression was in leukemia cell lines followed by prostate cancers and brain tumors”. This is not clear. Do the authors mean that leukemic cells in general had the highest levels of LSD1 expression? Based on Fig 3B, it would appear that the very high level of LSD1 expression in THP-1 cells would skew this, and that in fact levels of LSD1 expression are broadly similar.

13.  Page 9, Figure 3B. How many replicates were performed to generate the box plots?

14.  Page 9, line 337. The authors say they chose four AML cell lines for their profiling experiments. Jurkat cells are a T-cell line, not myeloid.

15.  Page 9, Table 2. Indicate in the table description that the listed IC50 values are in uM.

16.  Page 10, line 357. The data for the co-administration of GSK2879552 and ricolinostat is not shown in Fig S15. This must be added to the graphs.

17.  Page 10, section 3.4 “iDual increases cellular protein…...” How was the concentration of 4 uM for the inhibitors chosen? Based purely on the data shown in Table 2, ricolinostat is being used at 138% of the IC50, whilst iDual is being used at 25% of the IC50. Could the authors comment on this difference? This experiment should be repeated using drug concentrations that more closely match the respective IC50 values.

18.  Page 10, Fig 4B. This figure indicates an increase in acetylation of both H3K18 and H3K27 using iDual. However, the corresponding western blot, shown in Fig S16B, suggests that at best, iDual does not lead to any change in the levels of acetylation of these proteins, and in the case of H3K18, it actually looks like there is a decrease in H3K18Ac. How many replicates of these western blots were performed to generate the graph shown in Fig 4B? Or can the authors provide clarity on this discrepancy?

19.  Page 10, Fig 4D. Similar to comment 17, the drugs used in this experiment are all at different ratios based on their IC50. Again, this experiment should be repeated with drug concentrations that more closely match the respective IC50 values.

20.  Page 11, line 403. For consistency change “….were incubated with the dual inhibitor” to “…were incubated with iDual”.

21.  Figure S18 requires a figure legend.

22.  Figure S18 panel A and B. The x axis labels are incorrect. Presumably the label “2” in panel A should be “iDual, whilst “3” in panel B should be “iHDAC6”? These need to be corrected.

23.  There does not appear to a Fig S19 and S20. If this is truly the case, the supplementary figures need to be renumbered following Fig S18.

24.  Page 12, line 412. Change “dual inhibitor 2” to iDual for consistency.

25.  Page 13, line 444. “While doxorubicin alone at the sublethal concentration was only able to trigger mild apoptosis with 20% of cells in the apoptosis/dead quadrant, pre-treatment with iDual resulted in a dramatic response of over 80%.” This sentence can be misleading. Based on annexin V staining, the figure of 20% dead cells only measures cells in late apoptosis/dead quadrant. However, the 80% quoted in dual treated cells includes those cells in early apoptosis. If these cells are not included in the analysis, the figure is closer to 45%. However, if you measure Caspase3/7 activity, these values stated are correct. The authors need to indicate that these values are based solely on caspase activity.

26.  Page 13, line 446. As in the previous comment, the statement in this line is only true for the data shown for caspase 3/7. This should be made clear.

27.  Page 14, line 464. The authors state that pre-treatment of THP-1 cells with iHDAC6 did not lead to a significant increase in the level of gamma-H2AX when the cells were subsequently treated with doxorubicin. However, examination of Fig S21 shows that the levels of gamma-H2AX increases from 30.7% with dox alone, to 56.3% when the cells are pre-treated with iHDAC6. Was this increase not found to be significant? The authors need to comment further on this?

28.  Page 14, line 470. The authors say they examined the effects of doxorubicin intercalation into DNA. There is no data presented from these experiments.

29.  Page 15, line 482. The figure legend should relate to what is shown in the actual figure, not the corresponding supplementary figure. Please delete “...(Figure S21)...” and “…either…” and “…iHDAC6 or GSK2879552…” and “…or ricolinostat (1.25 μM, 72 hours)…”.

30.  Page 16, line 508. Change “...from...” to “...for…”

31.  Page 17, line 540. The authors conclude that the sensitization to doxorubicin is not due to any changes in the ability of doxorubicin to intercalate into DNA. However, as pointed out in comment 28, data is not presented to confirm this.

Round 2

Reviewer 2 Report

The authors have improved the quality of the manuscript in the new version after revision. Although they have not performed the appropriate tests on cells from AML patients, the results are clear, interesting and supported by data.

Reviewer 3 Report

The authors have improved the quality of the manuscript in the new version after revision. Although without proper control in terms of healthy bone marrow cells, results are supported by data.

Reviewer 4 Report

I would like to thank the authors for addressing my original comments. The changes that have been made further strengthen their manuscript